# Metabolic Disruption of Gold Nanospheres, Nanostars and Nanorods in Human Metastatic Prostate Cancer Cells

**DOI:** 10.3390/cells12050787

**Published:** 2023-03-02

**Authors:** Sílvia Soares, Cláudia Pereira, André P. Sousa, Ana Catarina Oliveira, Maria Goreti Sales, Miguel A. Correa-Duarte, Susana G. Guerreiro, Rúben Fernandes

**Affiliations:** 1BioMark@ISEP/CEB, Center of Biological Engineering of Minho University, School of Engineering, Polytechnic Institute of Porto, 4249-015 Porto, Portugal; 2FP-I3ID, Universidade Fernando Pessoa (UFP), 4249-004 Porto, Portugal; 3Instituto de Investigação e Inovação em Saúde (i3S), 4200-135 Porto, Portugal; 4Faculty of Chemistry, University of Vigo, 36310 Vigo, Spain; 5CEB—Centre of Biological Engineering of Minho University, 4710-057 Braga, Portugal; 6ICBAS—School of Medicine and Biomedical Sciences, University of Porto, 4050-313 Porto, Portugal; 7Faculty of Health Sciences (FCS) & Hosptal Escola Fernando Pessoa (HEFP), University Fernando Pessoa (UFP), 4249-004 Porto, Portugal; 8Biomark@UC/CEB, Centre of Biological Engineering of Minho University, Department of Chemical Engineering, Faculty of Sciences and Technology, Coimbra University, 3030-790 Coimbra, Portugal; 9CINBIO, University of Vigo, 36310 Vigo, Spain; 10Southern Galicia Institute of Health Research (IISGS), Biomedical Research Networking Center for Mental Health (CIBERSAM), 36310 Madrid, Spain; 11Institute of Molecular Pathology and Immunology of the University of Porto—IPATIMUP, 4200-465 Porto, Portugal; 12Department of Biomedicine, Biochemistry Unit, Faculty of Medicine, University of Porto, 4200-319 Porto, Portugal

**Keywords:** beta-oxidation, glycolysis, gluconeogenesis, gold nanoparticles, internalization, nanomedicine, reactive oxygen species

## Abstract

Nanomaterials offer a broad spectrum of applications in biomedicine. The shapes of gold nanoparticles could modulate tumor cell behavior. Spherical (AuNP_sp_), stars (AuNP_st_) and rods (AuNP_r_) shapes of polyethylene glycol coated-gold nanoparticles (AuNPs-PEG) were synthesized. Metabolic activity, cellular proliferation, and reactive oxygen species (ROS) were measured and the impact of AuNPs-PEG in metabolic enzymes function was evaluated by RT-qPCR in PC3, DU145, and LNCaP prostate cancer cells. All AuNPs were internalized, and the different morphologies of AuNPs showed to be an essential modulator of metabolic activity. For PC3 and DU145, the metabolic activity of AuNPs was found to rank in the following order from lowest to highest: AuNP_sp_-PEG, AuNP_st_-PEG, and AuNP_r_-PEG. Regarding LNCaP cells, the AuNP_st_-PEG were less toxic, followed by AuNP_sp_-PEG and AuNP_r_-PEG, but it seems not to be dose-dependent. The proliferation was lower in AuNP_r_-PEG in PC3 and DU145 cells but was stimulated around 10% in most conditions (0.001–0.1 mM) in LNCaP cells (not statistically significant). For 1 mM, LNCaP cells showed a significant decrease in proliferation only for AuNP_r_-PEG. The outcomes of the current study demonstrated that different AuNPs conformations influence cell behavior, and the correct size and shape must be chosen considering its final application in the field of nanomedicine.

## 1. Introduction

Nanomaterials have shown great promise in the fight against cancer, due to their unique properties and ability to interact with biological systems at the nanoscale [1]. In recent years, researchers have been exploring the use of nanomaterials in various cancer therapies and diagnostic tools, intending to improve the effectiveness and specificity of these treatments. Different nanoparticles can be used, including organic material (lipids, proteins, or polymers), hybrid (nanofoams) or inorganic material, such as metals or salts—gold, silver, magnetic. Although the use of gold for medical applications has a long history, there is an increasing interest in gold nanoparticles (AuNPs) in bioimaging and therapy of cancer [2]. Gold is the most stable noble metal, biocompatible, and its surface can be easily functionalized with various biomolecules [3,4]. Besides their unique chemical, optical and physical properties, AuNPs are easy to synthesize in different sizes (from 1 to 100 nm) and shapes (spheres, rods, stars, among others). The impact of AuNPs size has been extensively studied in the literature, but little is known about the AuNP shape effect in vitro. The spheres gold nanoparticles (AuNP_sp_) are the most known in the literature compared to rod gold nanoparticles (AuNP_r_) and star gold nanoparticles (AuNP_st_) or other shapes [2,5,6]. AuNPs have size- and shape-dependent physical and chemical properties [7]. In vitro studies have shown that AuNPs influence cellular uptake, cell-crosstalk, cell biodistribution, and optical properties [3,8,9]. However, their mechanism of action remains to be unveiled. Polyethylene glycol (PEG) was used as a surface coat of AuNPs to improve the monodispersity and their biocompatibility, to escape from the immune system surveillance [10].

Nanotechnology has been a trending area in medical applications, such as cancer. Chemotherapy and radiotherapy (RT) are two therapeutic approaches used against tumor cells but exhibited some constraints related to toxicity and treatment resistance [11]. Prostate cancer (PCa) is the most frequently diagnosed non-skin cancer and the leading cause of cancer death in men [12]. Consequently, patients with advanced or metastatic PCa do not have immediate and effective therapeutic interventions, presenting a 5-year survival rate of only 30% [13]. In addition, some patients do not respond to therapy protocols. Therefore, it is crucial further investigate new therapeutic strategies to target PCa.

So, to overcome the lack of knowledge regarding AuNPs in PCa, we used three different shapes of AuNPs-PEG (AuNP_sp_, AuNP_st_, and AuNP_r_) in three metastatic PCa cell lines with different origins (bone, brain, and lymph node). This study will enable us to understand the effects of AuNP shape in terms of their physical and biochemical characteristics in pathological conditions. To our knowledge, this study is the first to evaluate the effect of these three AuNPs in PCa metastatic cell behavior. The results will clarify which nanostructure(s) is the most suitable for metastatic PCa treatment.

## 2. Materials and Methods

### 2.1. Materials

Thiol-polyethylene glycol-amine (SH-PEG-NH_2_), molecular weight 2 kDa, trisodium citrate dehydrate (C_6_H_5_O_7_Na_3_·2H_2_O or NaCt), tetrachloroauric acid tetrahydrate (HAuCl_4_·4H_2_O, 99.99%), silver nitrate (AgNO_4_), sodium borohydride (NaBH_4_), hexadecyltrimethylammonium bromide (CTAB, ≥99%), L-ascorbic acid, ≥99%, fetal bovine serum (FBS), phosphate-buffered saline (PBS), trypsin-EDTA, and Antibiotic antimycotic solution from Sigma Aldrich^®®^ LLC., St. Louis, MO, USA; Roswell Park Memorial Institute (RPMI-1640) and Minimum Essential Medium (MEM) media were purchased from Biowest (Nuaillé, France); cell proliferation ELISA, BrdU kit and PrestoBlue^®®^ Cell Viability Reagent (PB) was obtained from Roche (Indianapolis, IN, USA) and Invitrogen Co. (San Diego, CA, USA) respectively. 2′,7′-Dichlorodihydrofluorescein diacetate (H_2_DCFDA) was acquired in Biotium (Hayward, CA, USA), QIAzol lysis reagent was purchased in QIAGEN Inc. (Valencia, CA, USA), EasyScript^®®^ Reverse transcriptase in Transgen Biotech Co., LTD. (Beijing, China) and RT-qPCR NZYSpeedy qPCR probe kit in NZYTech (Lisbon, Portugal).

### 2.2. Synthesis of AuNP

#### 2.2.1. Synthesis of AuNP_sp_-PEG

AuNP_sp_ were prepared according to Turkevich method and co-workers’ protocol using a HAuCl_4_·4H_2_O solution that is reduced and stabilized by trisodium citrate (NaCt) as illustrated in Figure 1A [14].

One hundred mL of 0.5 mM of HAuCl_4_·4H_2_O solution was prepared with ultrapure water in a triple-neck round-bottom flask and heated under vigorous stirring at 100 °C. Subsequently, 10 mL of 1% NaCt (*w*/*v*) was mixed with HAuCl_4_·4H_2_O solution. The solution was maintained for 15 min under heat until the red-wine colour was obtained. After, turn off the temperature and allow the solution to cool. After cooling, 1 mg/mL SH-PEG-NH_2_ was added to the AuNP_sp_ solution and incubated at 4 °C overnight. Then, AuNP_sp_-PEG were purified by centrifugation at 4500 rpm for 40 min, and the pellets were resuspended in ultrapure water and stored at 4 °C.

#### 2.2.2. Synthesis of AuNP_st_-PEG

AuNP_st_ were prepared according to the reported protocol of Tian and colleagues [15] as in Figure 1B. AuNP_st_ were first synthesized using a seed solution obtained by adding 3 mL of 1% NaCt (*w*/*v*) to 100 mL of 1.0 mM HAuCL_4_. Then, 100 µL of seed solution was added to 10 mL of 0.25 mM HAuCl_4_ at room temperature. Forty µL of 0.01 M AgNO_3_ and 50 µL of 0.1 M L-ascorbic acid were added. To coat PEG on the AuNP_st_ surfaces, 20 µL of SH-PWG-NH_2_ was added. The AuNP_st_-PEG were collated by centrifugation at 5200 rpm and redispersed in water.

#### 2.2.3. Synthesis of AuNP_r_-PEG

First, AuNP_r_ seeds were prepared by mixing 25 µL of 50 mM HAuCl_4_ with 4.7 mL of 0.1 M CTAB solution in a water bath at 27–30 °C (Figure 1C). Next, 300 µL of 10 mM NaBH_4_ solution was added to the previous solution under constant stirring. To synthesize AuNPs, a seed growth solution was prepared based on Scarabelli and co-workers [16]. Ten mL of 100 mM CTAB were incubated with 100 µL of 50 mM HAuCl_4_ under gentle stirring. Then, 75 µL of 100 mM L-ascorbic acid was added to the mixture for a few seconds. Eighty µL of 5 mM of AgNO_3_ was added to the growth solution for a few seconds. Finally, 120 µL of seeds solution was added to the previous mixture and left undisturbed at 27 °C for 30 min. To remove the excess solution reagents, AuNP_r_ was centrifugated twice at 7500 rpm for 30 min. The next step was PEGylation by adding 0.2 mM of SH-PEG-NH_2_ to the AuNP_r_ solution. After stirring for 24 h, the solution was washed twice at 7500 rpm for 30 min.

### 2.3. Characterization Methods of AuNPs

#### 2.3.1. UV-Visible

The UV-Visible (UV-Vis) absorption spectra of different solutions were measured in a 1 mm quartz cuvette at room temperature using an Evolution 200 Series spectrophotometer (Thermo Fisher Scientific, Waltham, MA, USA). The absorption values were used to determine the concentration of particles in the solution.

#### 2.3.2. Transmission Electron Microscopy

The size and morphology of the samples were investigated using transmission electron microscopy (TEM). Ten µL of each sample was mounted on Formvar/carbon film-coated mesh nickel grids (Electron Microscopy Sciences, Hatfield, PA, USA). For experiments with PEG, prepared samples were contrasted with 10 µL of phosphotungstic acid (PTA) and placed on the grid. After, grids were observed in a JEM 1400 TEM (JOEL Ltd., Tokyo, Japan) with an accelerating voltage of 80 kV. Images were digitally recorded using a CCD digital camera Orious 1100 W Tokyo, Japan, and analysed using ImageJ software to create a size histogram based on representative images obtained.

#### 2.3.3. Scanning Electron Microscope

A scanning electron microscope (SEM) was used to confirm nanoparticle production and examine nanoparticle morphology. Ten µL of samples were deposited onto silicon wafers and left undisturbed until evaporating the solvent at room temperature. SEM images were acquired using a FEI Quanta 400 FEG ESEM/EDAX PEGASUS X4M equipment.

#### 2.3.4. Dynamic Light Scattering and Zeta Potential

Also, nanoparticles’ hydrodynamic diameter and zeta potential were measured by dynamic light scattering (DLS) using a Zeta Sizer Malvern Nano series (Malvern Instruments Ltd., Malvern, UK). All average particle sizes reported here are based on scattered light intensity weighted averages. Five DLS measurements were made for each sample suspension with a fixed run time of 30 s. The scattering/detection angle was set at 173°.

### 2.4. AuNPs-Cells In Vitro Assays

#### 2.4.1. Cell Culture

The PCa cell lines used in this study were PC3 (ATCC^®®^ CRL-1435™), LNCaP (ATCC^®®^ CRL-1740™), and DU145 (ATCC^®®^ HTB-81™) cells. PC3 and LNCaP cells used RPMI-1640 medium, and DU145 cells used MEM medium. Culture media were supplemented with 10% FBS and 1% penicillin/streptomycin [17,18,19]. Cells were maintained in culture at 37 °C with 5% CO_2_. For these experiments, cells were used between 8 and 15 passages.

Cell lines were cultured and grown to ~80% confluence and sub-cultured for different assays. Cells (1 × 10^5^ cells/well) were cultured in 96-well plates (VWR) for 24 h. Then, cells were washed with PBS and treated with AuNPs for 24 h at 37 °C with 5% CO_2_ in a humidified environment. Different concentrations of AuNP_sp_, AuNP_st_ and AuNP_r_ ranging from 0 to 1 mM were prepared in serum-free conditions.

#### 2.4.2. Qualitative Analysis of the Cellular Uptake of AuNPs

To evaluate the cellular uptake of different concentrations of AuNPs using TEM images and flow cytometry. Cells were treated with different AuNPs solutions and incubated for 24 h. Then, the cells were washed, trypsinized and resuspended in TEM fix solution (2.5% glutaraldehyde and 2% paraformaldehyde in 0.1 M sodium cacodylate) for three days. After, the fix solution was removed, and cells were washed in 0.1 M sodium cacodylate buffer. Next, a post-fix solution (2% osmium tetroxide in 0.1 M sodium cacodylate) was added to the samples. After 2 h, the samples were washed and centrifuged three times in water. Then, they were incubated with 1% Uranyl acetate for 30 min. The pellet was then included in Histogel^TM^ (Thermo Fisher Scientific, Waltham, MA, USA, HG-4000-012). Finally, the samples were dehydrated in a graded series of ethanol solutions (50%, 70%, 80%, and 100%) and treated with propylene oxide (3×). Ultrathin sections of the samples were cut and observed with a JEM 1400 TEM (JOEL Ltd., Tokyo, Japan) with a CCD digital camera Orious 1100 W Tokyo, Japan. Then, the intracellular location of the AuNPs was analysed.

Regarding flow cytometry, cells (1 × 10^6^) were plated in 6-well and treated with each AuNPs solutions. The next day, the solution was removed, the cells were washed, and then the cells were collected using trypsin. Cells were examined using ATTUNE flow cytometer (Thermo Fisher Scientific, Waltham, MA, USA).

#### 2.4.3. Cellular Viability

Viable cells can metabolize resazurin into resofurin on mitochondria [20]. Cells were incubated with AuNP treatments for 24 h. Afterwards, 10 µL of resazurin was added directly into 90 µL of culture medium. Upon incubation for 1 h at 37 °C, 100 μL/well was transferred to a new 96-well plate. The absorbance was measured using a Spectra Max Gemini XS (Molecular Devices, San Jose, CA, USA) at excitation and emission wavelengths of 550 and 600 nm, respectively.

#### 2.4.4. Cellular Proliferation

After 24 h of treatment, the cells were incubated with BrdU solution at a fifinal concentration of 100 µM for 2 h. The cell proliferation assay was performed according to the manufacturer’s instructions [21]. The results were expressed as a percentage of control (100%) and tested in duplicates on two independent experiments.

#### 2.4.5. Reactive Oxygen Species (ROS)

Molecular probe 2′,7′-Dichlorodihydrofluorescein diacetate (H2DCFDA) assay was dissolved in dimethyl sulfoxide (DMSO) at 10 mM stock solution. After plating cells, adherent cells were washed with buffer and stained with a 10 µM probe for 45 min at 37 °C in the dark. Next, cells were rewashed and were treated with AuNPs for 24 h. Cells were then analysed on a fluorescence plate reader (SpectraMax^®®^ Gemini™ EM Microplate Reader, Molecular Devices, San Jose, CA, USA) at excitation/emission of 504/529 nm in endpoint mode.

#### 2.4.6. RNA Isolation and Gene Expression

Cells (4–8 × 10^5^ cells/well) were seeded in 6-well culture plates and grown overnight. Then, cells were treated with different concentrations of 0.1 mM of AuNPs for 24 h. Total RNA was isolated from different types of samples followed QIAzol (Qiagen, Crawley, UK). The amount of DNA and RNA was determined using a Thermo Scientific™ Multiskan SkyHigh Microplate spectrophotometer (Life Technologies Fisher Scientific, Waltham, MA, USA). The ratio of absorbance at 260 nm and 280 nm was used to assess the purity of DNA and RNA. RNA was reversely transcribed using EasyScript^®®^ Reverse transcriptase (Transgen biotech, Beijing, China) and following manufacturer recommendations. RNA was subjected to RT-qPCR (NZYSpeedy qPCR probe kit, NZYTech, Lisbon, Portugal) using primer sets specific to hexokinase-2 (HK2), glucose-6-phosphatase (G6Pase), pyruvate kinase (PKM), pyruvate carboxylase (PCX), acyl-CoA dehydrogenase (ACADS) and mitochondrial fission 1 protein (FIS1, Table 1).

Threshold cycle (CT) values from each sample were plotted with two experimental replicates following the manufacturer’s procedure. The melting curve analysis was used to monitor the specificity of primers and probes. The expression level of each gene was normalized to the expression of the GAPDH housekeeping gene, and gene relative expression was employed by the ΔCT expression/ΔCT control ratio.

### 2.5. Statistical Analysis

All data are presented in mean ± standard deviation (SD) of experiments repeated at least three times. Data were analysed through Prism 8.0 (GraphPad Software, Boston, CA, USA). Differences between treatments were evaluated by two-way ANOVA with Sidak multiple comparisons test, according to the number of conditions and treatments. Results were considered significant when *p* < 0.05.

## 3. Results

### 3.1. Characterization of Different Shapes of AuNP

AuNPs with different conformations (AuNP_sp_, AuNP_st_, AuNP_r_) were used to compare their chemical, physical and biological effects. All AuNP conformations were functionalized with PEG to improve cellular uptake and overcome the immune system as described in the literature. AuNPs exhibited different surface plasmon resonance (SPR) bands in UV–Vis absorption spectra over 400–1000 nm, as shown in Figure 2D–F.

AuNPs’ size and shape were observed by TEM and SEM analyses, respectively—Figure 2G–L. The average size for AuNP_sp_-PEG was 18.4 ± 2.1 nm, and a SPR peak was about ~522.3 nm. For AuNP_st_-PEG, the average size was 80.7 ± 18.9 nm, and a broad plasmon band mainly ranging from 480 nm to 1000 nm with a maximum at ~906.3 nm was observed. AuNP_r_-PEG were synthesized using the seed-mediated method to obtain 45.4 ± 4.5 nm × 11.6 ± 1.2 nm (length × width) by TEM (with an aspect ratio of around 3.9:1) and exhibit a dominant longitudinal SPR peak of ~763.6 nm and a minor transverse peak at ~513.6 nm. From the UV-Vis spectra, TEM, and SEM images, AuNP_sp_-PEG, AuNP_st_-PEG, and AuNP_r_ -PEG had spherical, star and rod structures matching their designs. Finally, a histogram size was created using TEM images where over 50 particles were counted—Figure 2M–O.

From DLS (Table 2), the average size for AuNP_sp_-PEG was about 47.31 ± 0.46 nm, AuNP_st_-PEG was 109.61 ± 1.27 nm, and AuNP_r_ -PEG was 54.58 ± 0.34 nm.

These AuNPs hydrodynamic size values were different from the ones obtained in TEM analysis, because on DLS the PEG chains layer was hydrated on the surface of nanoparticles [22]. According to the polydispersity index (PDI) of AuNPs, the AuNP_st_-PEG exhibited more monodispersity than AuNP_sp_-PEG and AuNP_r_-PEG. In addition, the zeta potential measurement demonstrated that AuNPs were successfully conjugated with PEG and all nanostructures were positively charged. AuNP_sp_-PEG, AuNP_st_-PEG, and AuNP_r_-PEG indicated a zeta potential of 6.7 ± 7.9, 33.1 ± 12.0, and 11.0 ± 18.9 mV, respectively.

### 3.2. Qualitative Analysis of the Cellular Uptake of AuNPs-PEG

Cellular uptake of AuNPs-PEG involves highly regulated mechanisms with biomolecular interactions: shape, size, and capping dependents [23]. Also, AuNPs have multiple different cellular entry routes to cross the cell plasma membrane, including passive translocation across the cell membrane or through active endocytosis [23,24,25]. In the present study, cells were treated for 24 h with different structures of AuNPs at a concentration of 0.01 mM prior to TEM analysis to investigate cellular internalization. We performed a qualitative analysis of the cellular uptake of AuNPs using TEM images, and they revealed numerous high electron density-staining particles inside the cells incubated with AuNPs (Figure 3). AuNPs-PEG were not found in control groups (Figure 3A–C), whilst an interesting morphological phenomenon was found in treated groups.

The three metastatic cell lines internalized the AuNP_sp_-PEG, AuNP_st_-PEG and AuNP_r_-PEG. TEM images showed AuNPs clusters distributed across the cytoplasm. Most AuNPs-PEG are trapped inside the endosome’s vesicles, most of which are in the proximity of mitochondria and the endoplasmic reticulum. The cell nuclei do not seem to be affected by AuNPs-PEG. TEM data demonstrated the cellular uptake of AuNPs in the three cell lines. Qualitatively, AuNP_st_-PEG appears to be more extensively accumulated than AuNP_sp_-PEG and AuNP_r_-PEG.

Another complementary analysis was performed by flow cytometry using the forward-scattered light (FSC), proportional to the cell size and the side-scattered light (SSC) related to cell’s internal complexity. Results showed that after 24 h of incubation with AuNPs-PEG (Figure 4), the uptake was higher in case of AuNP_sp_-PEG following AuNP_st_-PEG and AuNP_r_-PEG in all cell lines.

However, for DU145 cells, only some minor changes were found in AuNP_sp_-PEG and AuNP_st_-PEG. For LNCaP, modifications on complexity were identified only for AuNP_sp_-PEG.

### 3.3. AuNPs Decrease Prostate Cancer Cells Viability

A broad spectrum of particle concentrations was tested to investigate the biological effect of AuNPs-PEG on cell viability of metastatic PCa cell lines—Figure 5A–C [26,27,28].

After 24 h treatments, all metastatic cell lines showed a reduction of cell viability compared to the control (cells without AuNP treatment). The results demonstrated that the cellular viability is independent of AuNPs concentration. PC3 and DU145 cells viability was between 50–100% compared to control upon treatment with 0.001 to 1 mM of AuNP_sp_-PEG or AuNP_st_-PEG. When treated with 0.001 to 0.1 mM of AuNP_r_-PEG, PC3 and DU145 cells viability was 70–80%. However, 1 mM AuNP_r_-PEG treatment revealed a higher decrease in cellular viability on PC3 and DU145 cell lines (52.5% and 52.9%, respectively for PC3 and DU145, *p* < 0.05). In the case of LNCaP, all treatments of AuNPs-PEG with different concentrations decreased the cellular viability.

### 3.4. AuNPs Modulate Prostate Cancer Cell Proliferation

The cellular proliferation was performed using the BrdU cell assay—Figure 5D–F. When PC3 and Du145 cells were treated with 0.001–0.1 mM AuNPs-PEG concentrations of each shape, cell proliferation rate decreased compared to controls. Contrariwise, on LNCaP cells, the same treatments of AuNPs-PEG did not reveal a statistically significant difference in cell proliferation after 1 mM of AuNP_r_-PEG treatment (*p* < 0.001).

### 3.5. Cellular Internalization of AuNPs

TEM analysis has shown that all shapes of AuNPs-PEG can be internalized by PC3, DU145 and LNCaP cells and created ultrastructure changes. An increase in vacuolization and numerous autophagic vacuoles in the three cell lines were observed by TEM (Figure 3).

### 3.6. Intracellular ROS Levels Depend on AuNPs-PEG Shape Treatment

Cells were treated with 0.1 mM of different shapes of AuNPs-PEG for 24 h, and then ROS levels were observed (Figure 6).

AuNP_sp_-PEG decrease ROS levels when compared to control group in PC3 and DU145 cells. Remarkably, treatment with 0.1 mM of AuNP_r_-PEG only decreased ROS levels on DU145 cells. In LNCaP cells, the treatments did not alter the ROS levels when compared to the control group (*p* > 0.05).

### 3.7. AuNPs-PEG Shape Affects Mitochondria Biogenesis and Metabolic Function

Changes in metabolic function can contribute to the growth and progression of PCa. Understanding these changes in metabolic function may provide new targets for the development of PCa therapies. So, the impact of different AuNPs-PEG in the expression of enzymes involved in metabolic pathways, such as HK2, G6Pase, PKM, PCX, and ACADS was evaluated (Figure 7 and Figure 8).

Besides, mitochondria are highly dynamic organelles in cancer biology and are a crucial player on the altered cancer energy metabolism. To investigate the effect of AuNPs-PEG treatment on cancer cell energy metabolism, FIS1 mRNA levels, a critical checkpoint for mitochondria division involved in the genetic regulation of several metabolic pathways, such us, glycolysis, gluconeogenesis, and beta-oxidation was determined (Figure 7 and Figure 8).

PC3 cells treated with AuNP_sp_-PEG and AuNP_r_-PEG presented an increase of mRNA expression of HK2 and a decrease of PKM, involved in the first and the last step of glycolysis, respectively. On the other hand, DU145 cells and LNCaP cells did not have statistically significant differences in these transcripts. Gluconeogenesis is another metabolic pathway that fully occurs in hepatocytes. All three cell lines express PCX and G5Pase mRNA, encoding the first and final gluconeogenesis step. PC3 cells treated with AuNP_sp_-PEG and AuNP_r_-PEG presented increased mRNA expression of these two mRNA enzymes. DU145 and LNCaP cells did not have statistically significant differences in gluconeogenesis mRNA expression genes upon ant treatment.

Fatty acids and glucose can be used by the cells as energy sources through beta-oxidation and glycolysis pathways, respectively, resulting in acetyl-CoA. If acetyl-CoA increases, FIS1 ubiquitination can occur, decreasing mitochondria fission. On PC3 cells treated with AuNP_sp_-PEG and AuNP_r_-PEG, an increase of ACADS and FIS1 mRNA expression was determined. DU145 cells did not show statistically significant differences regarding enzymes expression for any treatment. However, a tendency to increased FIS1 was observed after AuNP_r_-PEG treatment. AuNP_sp_-PEG and AuNP_st_-PEG treatment increased the expression of ACADS mRNA in LNCAPs. No statistically significant differences were observed for FIS1 gene expression.

## 4. Discussion

Distinct methods were used to characterize the mean size of AuNPs-PEG, like TEM and DLS. The shape of AuNPs were confirmed by UV-Vis spectra, TEM and SEM image analysis. Considering particle size, data obtained from DLS measurement are usually bigger than those obtained from TEM due to the presence of the PEG chain and the layer hydration around the AuNPs solution [29,30]. Our synthesis process is in accordance with the literature, by the applied synthesis methods [14,15,16]. In our case, it was possible to characterize the three AuNPs-PEG with DLS. Still, by applying other techniques, such as depolarized dynamic light scattering (DDLS), it is possible to obtain results for specific anisotropic nanoparticles more similar to TEM results [31]. Regarding AuNP_r_-PEG, DLS measurements can provide a reasonably hydrodynamic diameter, which can be related to the length of AuNP_r_-PEG [32].

Regarding the shape, AuNP_sp_-PEG presented only one peak, AuNP_r_-PEG showed two peaks, and AuNP_st_-PEG exhibited a broad absorption band, which can be derived from the high density of surface spikes [22]. So, UV-vis showed different absorption patterns depending on the geometries, which agrees with the literature [26,33]. Additionally, the different morphologies were confirmed by TEM and SEM images. The validation of AuNPs-PEG was confirmed by positive values in zeta potential, increasing the stability of nanostructures, mainly AuNP_st_-PEG. The surface of AuNPs can be modified with several materials. Still, PEG is one of the biocompatible polymers most used in biomedicine because it improves the stability, internalization, and absorption of the AuNPs inside the cell. Besides, PEG contributes to reduced immunogenicity and elimination by clearance of AuNPs, increasing their circulation time in blood [28,34]. Also, PEG reduced the toxicity of AuNPs and improved their accumulation in tumor cells via the enhanced permeability and retention (EPR) effect [35,36,37]. Furthermore, Fytianos, et al. demonstrated that the cellular uptake of AuNPs modified with PEG-NH_2_ was higher than other functionalized surfaces, such as carboxylic acid—PEG-COOH [38].

Only a few publications analyzed the shape of AuNPs-PEG as an essential modulator of cytotoxicity, although extensive knowledge about AuNP’s cytotoxicity has been gathered. Our study allows evaluating at the same time different shapes of AuNPs-PEG using a concentration range to treat three metastatic cell lines of PCa [39]. These cell lines, PC3, DU145 and LNCaP, originated from different metastases of PCa, bone, brain, and supraclavicular lymph node, respectively [40]. Also, the LNCaP cell line is responsive to androgen and produces prostate-specific antigens (PSA). DU145 and PC3 cell lines are androgen-independent and have moderate and high metastatic potential, respectively [41,42,43]. Thus, analyzing different cell lines with other features, such as aggressiveness and hormonal dependence, provide a holistic overview of a wide range of PCa [42].

The uptake of different conformations of AuNPs-PEG by these three cell lines was analysed. Cells were treated with 0.01 mM AuNPs-PEG for 24 h. TEM findings revealed that all shapes of AuNPs-PEG suffered endocytosis in PC3, DU145, and LNCaP cells. We confirmed that AuNPs-PEG might be internalized by endosomes and vesicular bodies into PCa cells, as previously described [33,44,45].

AuNP_st_-PEG is the more captured nanoparticles by cells, appearing in clusters in all cell lines studied. They were detected in vesicles after 24 h of incubation. Remarkably, AuNP_sp_-PEG and AuNP_r_-PEG were found in sections after 24 h of incubation in all cell lines, but in less amount than AuNP_st_-PEG. It was demonstrated that citrate AuNP_sp_ has a better internalization capacity when compared with AuNP_r_ stabilized by citric acid ligands because AuNP_sp_ has less contact area with cell membrane receptors, increasing the number of NPs that can be internalized in Hela cells [9]. Similarly, Lee and co-workers compared chitosan-capped AuNP_sp_, AuNP_st_, and AuNP_r_ synthesized using green tea extract and concluded that AuNP_sp_ exhibited the fastest internalization rate than other shapes (AuNP_sp_ > AuNP_r_ > AuNP_st_) and lower toxicity in human hepatocyte carcinoma cells HepG2 [8]. However, to better understand the shape effect of the AuNPs on cell interaction, more studies should be developed to contribute to more efficient therapeutic nanosystems, reducing the therapeutic resistance related to conventional treatments.

In addition, our results showed a tendency to decrease the metabolic activity with increased concentration in AuNP_sp_-PEG, AuNP_st_-PEG, and AuNP_r_-PEG. Also, AuNP_r_-PEG showed a more significant decrease in metabolic activity than AuNP_sp_-PEG and AuNP_st_-PEG. The results are comparable to other outcomes of cytotoxicity in a similar range of concentrations, and the 0.1 mM concentration seems to be the safe dose of AuNPs-PEG [46,47]. LNCaP cells were not so sensitive, slightly reducing the viability and enhancing cell proliferation at the highest concentration compared to the other cell lines. This result can be due to their low growth rate observed by us and others [18]. In general, this study demonstrated that distinct morphologies have different cellular metabolic effects that can be caused by two factors—size or shape. Besides that, the results also suggest that AuNPs-PEG influence mitochondria functioning because using PrestoBlue^®®^ assay showed their cytotoxicity.

Moreover, it is known that cell cytotoxicity of AuNPs depends on the concentration used and the duration of the treatment [33]. Our findings indicated that cells respond in different manners to AuNPs treatment.

Additionally, TEM images exhibited a loss of integrity of cellular membranes and morphological differences of mitochondria, showing a higher number of mitochondria and more condensed ones. Moreover, disruption of the cell membrane, oxidative stress, cytoskeleton destruction, autophagy, and lysosomal dysfunction are essential functions and potential explanations for the cytotoxicity of AuNPs [8]. More studies should be done to analyze the detailed mechanisms of the cytotoxicity effect. Ultimately, the decreased metabolic activity is likely related to the harmful effect of aggregates, as suggested by Connor et al. [48].

Based on the literature and as we mentioned before, metabolic activity can be influenced by several factors that difficult the comparison between studies, such as shape, size, physicochemical surface properties, concentration, exposure time, cell type, experimental design and implementation, and analytical methods, because of variety of bioapplication of AuNPs [43,49]. According to our knowledge, it is the first study comparing the cytotoxicity of different morphologies of AuNPs-PEG in three distinct metastatic PCa cell lines. Nevertheless, Favi et al. showed that AuNP_sp_ (61.46 ± 4.28 nm) were more cytotoxic than AuNP_st_ (33.69 ± 8.45 nm) in human skin fibroblasts and fat rat pad endothelial cells (RFPECs) [50]. Also, another study compared AuNP_sp_ (~61.06 nm), and AuNP_r_ (534 nm × 65 nm) negatively charged and concluded that AuNP_sp_ presented more significant toxicity than AuNP_r_ in fibroblast cells [51]. Tarantola and co-workers showed that AuNP_sp_ (43 ± 4 nm) was more cytotoxic than AuNP_r_ (38 ± 7 nm × 17 ± 3 nm) with identical surface functionalization with CTAB in MDCK II cells, and the authors related the cytotoxicity to a higher release of toxic CTAB upon intracellular aggregation [45]. Woźniak et al. compared different AuNPs size and shapes (AuNP_sp_, ~10 nm, nanoflowers, ~370 nm, AuNP_r_, ~41 nm, nanoprisms, ~160 nm, and AuNP_st_, ~240 nm) in both HeLa and human embryonic kidney cells (HEK293T) cell lines and showed that AuNP_sp_ and AuNP_r_ were more cytotoxic than nanoflowers, nanoprisms, and AuNP_st_ [33]. One more time, the authors suggested that the tiny size of AuNPs and the aggregation process can influence the cytotoxicity of AuNP_sp_ and AuNP_r_. More recently, Steckiewicz et al. compared the cytotoxicity of AuNP_r_ (~39 nm × 18 nm), AuN_st_ (~215 nm), and AuNP_sp_ (~6.3 nm) in human fetal osteoblast (hFOB 1.19), osteosarcoma (143B, MG63) and pancreatic duct cell (hTERT-HPNE) lines. They showed that the cytotoxicity of AuNPs was shape-dependent, and AuNP_st_ were the most cytotoxic against human cells, followed by AuNP_r_ and AuNP_sp_ [44].

Besides the biosafety and toxicity of AuNPs, there is a gap regarding the molecular mechanisms and factors that influence nanomaterial toxicity. Researchers have found that AuNPs could affect the expression of intracellular metabolites and consequently change the expression of the functional genome, transcriptome, and proteome [52,53,54]. Thus, metabolic reprogramming of tumor cells has emerged as a new therapeutic strategy. After the Warburg effect, where oxidative phosphorylation in proliferative cells was switched to glycolysis even in aerobic conditions, the metabolic changes in tumor cells began to be explored [55]. The sensitivity of tumor cells reveals different sensitivities to various molecules related to gluconeogenesis, glycolysis, or fatty acid synthesis pathway [56]. Although some studies explored the effect of AuNPs on tumor cell metabolism, is still a lot to uncover [57,58,59].

PC3 cells treated with AuNP_sp_-PEG and AuNP_r_-PEG presented increased gene expression involved in cell replication (Figure 7 and Figure 8). AuNP_st_-PEG triggers a global reduction in cellular metabolism and activity. DU145 cells treated with AuNP_sp_-PEG and AuNP_st_-PEG inactivate the whole central cell metabolism, as reflected in the decrease in cell viability, glycolytic pathways, oxidation of fatty acids and mitochondrial replication. Cells treated with AuNP_r_-PEG also showed increased mRNA expression of most enzymes implicated in energy metabolism. In LNCaP cells, AuNP_sp_-PEG prompted the reduction of gluconeogenesis enzymes and glycolytic enzyme HK2. However, there is an increased expression of beta-oxidation ACADS enzyme and an increase in PKM expression, resulting in increased acetyl-CoA concentrations that enter the TCA cycle. There was also a reduction in FIS1 mRNA, implying mitochondrial metabolic activity reduction. Treatment with AuNP_st_-PEG resulted in the upregulation of enzymes involved in glycolysis, beta-oxidation, and gluconeogenesis, suggesting the induction of energy metabolism and anabolic pathways required for proliferative cell activity. Furthermore, treatment with AuNP_r_-PEG led to FIS1 gene downregulation. Given that FIS1 is involved in mitochondrial replication, these findings led to the assumption that AuNP_r_-PEG induces cell metabolism inactivation. Additionally, AuNP_r_-PEG presented a slight stimulation of the first step of glycolysis and an inhibition of beta-oxidation. In general, AuNP_sp_-PEG and AuNP_r_-PEG tend to increase the expression of enzymes involved in glycolysis, such as HK2 and PKM in PC3 and LNCaP cells, suggesting they play a role in supporting cancer cell survival. Also, AuNPs slightly increased G6Pase in the PC3 cell line. It can be hypothesized that AuNPs may promote NADPH production, which plays a role in reductive synthesis (e.g., lipogenesis and cholesterol) and is a key regulator of the antioxidant defense. Overall, the effect of AuNPs on the expression of metabolic enzymes is complex and context dependent. While AuNPs may disrupt the energy production and biosynthesis pathways in cancer cells, they may also promote the production of NADPH and support cancer cells’ survival. Further studies are needed to fully understand the mechanisms behind the effects of AuNPs on metabolic enzymes and their potential implications for cancer therapy.

## 5. Conclusions

Clinical development of treatments or therapeutic agents is essential to support an optimal management strategy for this challenging disease, the PCa.

Until now, this was the first study to compare the cytotoxicity of different morphologies of AuNPs and to evaluate the effect of different AuNPs-PEG on cellular metabolic enzyme levels in three distinct metastatic PCa cell lines. The analysis of cellular metabolism should be considered to ensure safety is preserved whenever AuNPs are applied in the clinic. This study demonstrated that distinct morphologies of AuNPs influenced the metabolic activity in these three cell lines evaluated, being a potential modulator of cell viability, proliferation, and metabolic enzymes. Also, our study showed that AuNPs are concentration-dependent and cell-type-dependent. For PC3 and DU145, AuNP_sp_-PEG were less toxic, followed by AuNP_st_-PEG and AuNP_r_-PEG. We observed that for LNCaP cells, the AuNP_st_-PEG were the less toxic, followed by AuNP_r_-PEG and AuNP_sp_-PEG. In general, the AuNP_r_ seem to be dose-dependent and the most efficient shape to destroy these two types of tumour cells with statistically significant results. Additional studies must be performed to properly quantify the cellular uptake efficiency of AuNPs and understand the effect of size and shape singly. After evaluating the effect of AuNPs on cell metabolism, AuNP_sp_ showed opposite results between PC3 and DU145. We believe that the surface markers activated in each cell line differ due to the different membrane compositions.

Regarding the effect of AuNP_st_-PEG and AuNP_r_-PEG, they seem to cause similar responses in more aggressive lines (PC3 and DU145) and to inactivate cell metabolism in more sensitive lines, such as LNCaP. So, this diverse response observed may be related to the different cell line characteristics, namely expressed markers on the membrane, and androgen receptor dependence, among others. However, more studies should be done to understand the mechanisms behind these differences.

## Figures and Tables

**Figure 1 cells-12-00787-f001:**
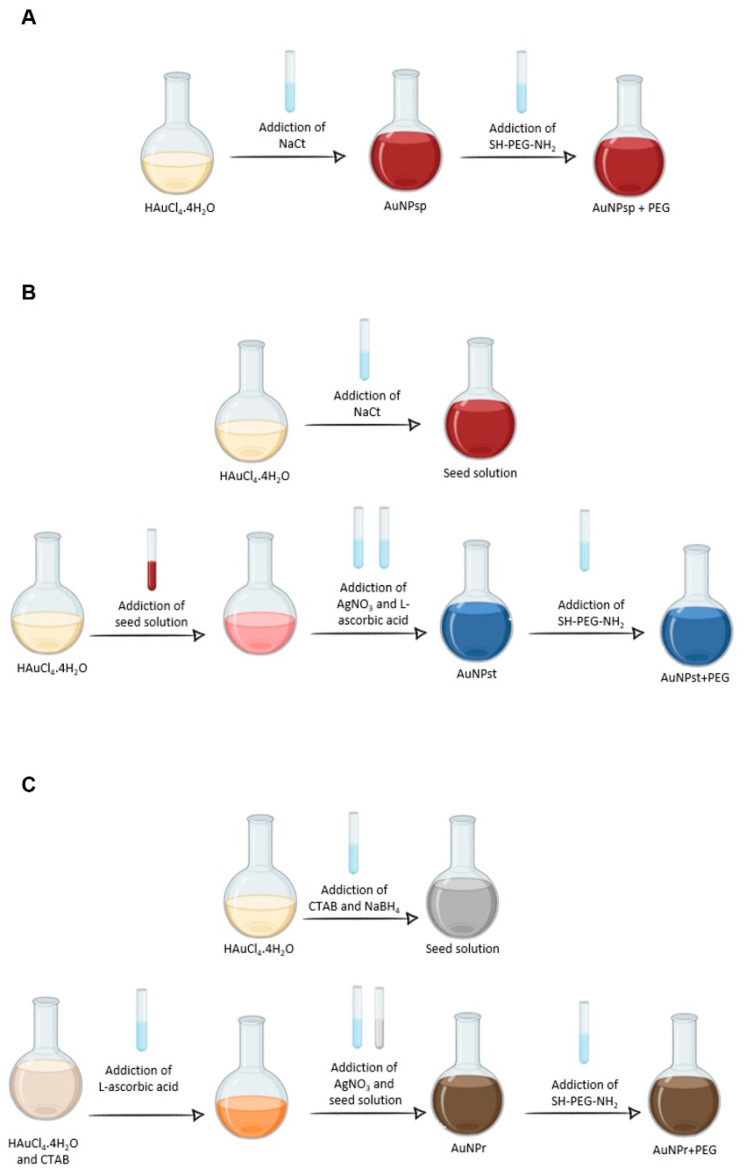
Schematic diagram of the synthesis process of pegylated gold nanoparticles: (**A**) spherical gold nanoparticles (AuNP_sp_); (**B**) gold nanostars (AuNP_st_), and (**C**) gold nanorods (AuNP_r_). HAuCl_4_·4H_2_O, tetrachloroauric acid tetrahydrate (99.99%); NaCt, trisodium citrate dehydrate; SH-PEG-NH_2_, thiol-polyethylene glycol-amine; AgNO_4_, silver nitrate; NaBH_4_, sodium borohydride; CTAB, hexadecyltrimethylammonium bromide ≥99%; L-ascorbic acid, ≥99%.

**Figure 2 cells-12-00787-f002:**
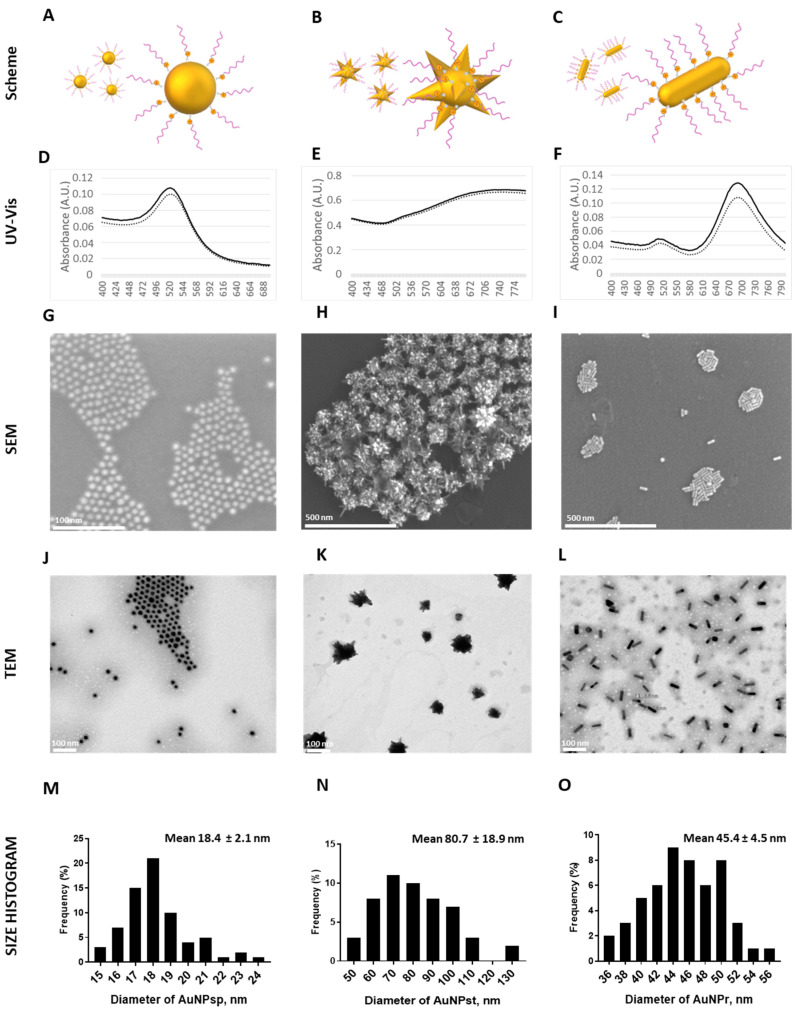
Characterization of PEG-coated AuNPs with different conformations of gold nanoparticles. Representative illustration of PEG-coated spherical gold nanoparticles (AuNP_sp_-PEG, (**A**); PEG-coated star gold nanoparticles (AuNP_st_-PEG, (**B**) and PEG-coated rod gold nanoparticles (AuNP_r_-PEG, (**C**). Characterization by UV-Vis spectra, results of AuNP_sp_ (**D**), AuNP_st_ (**E**) and AuNP_r_ (**F**). The dash line means PEG-coated AuNPs, and solid line means AuNPs without PEG. Characterization by SEM of AuNP_sp_-PEG image scale of 100 nm (**G**), AuNP_st_-PEG scale of 500 nm (**H**) and AuNP_r_-PEG scale of 500 nm (**I**). Characterization by TEM of AuNP_sp_-PEG (**J**), AuNP_st_-PEG (**K**) and AuNP_r_-PEG (**L**) with PEG staining with phosphotungstic acid, image scale 100 nm. Histogram size based on TEM images of AuNP_sp_-PEG (**M**), AuNP_st_-PEG (**N**) and AuNP_r_-PEG (**O**), the scale of 100 nm. The histogram size was obtained by counting over 50 particles.

**Figure 3 cells-12-00787-f003:**
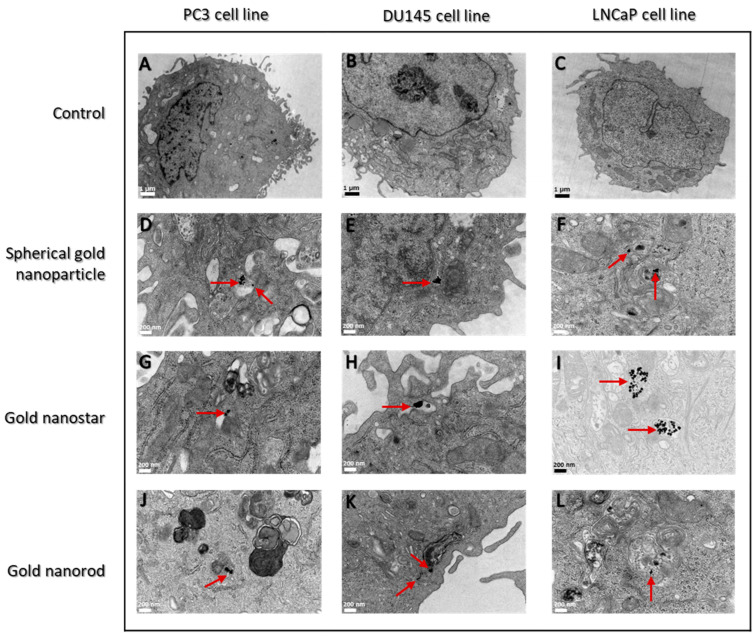
Results of AuNPs-PEG uptake by prostate cancer cell lines (PC3, DU145 and LNCaP) treated with 0.01 mM AuNPs for 24 h. Representative TEM images of ultrathin sections. Red arrows, indicate the presence of AuNPs inside all cell lines. Images of cells without AuNPs-PEG treatment (control, (**A**–**C**)). Cells treated with AuNP_sp_-PEG (**D**–**F**); cells treated with AuNP_st_-PEG (**G**–**I**); and cells treated with AuNP_r_-PEG (**J**–**L**). TEM magnification image 12,000× (**A**–**C**) and 50,000× (**D**–**L**). SEM scale bar is 1 µm to control cells and 200 nm for treated cells.

**Figure 4 cells-12-00787-f004:**
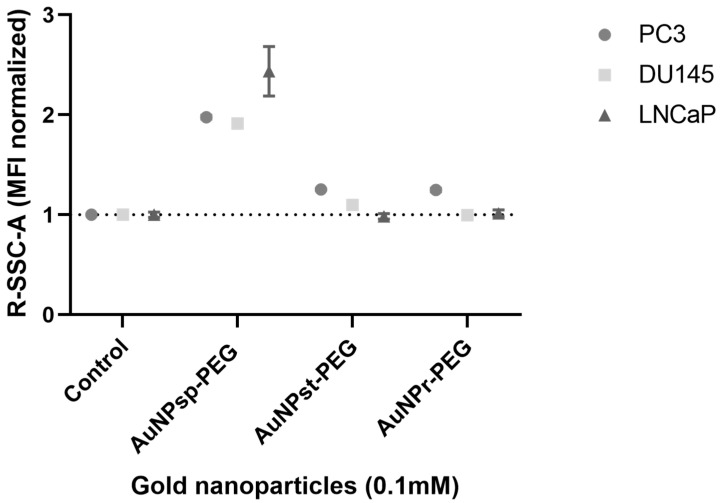
Cellular complexity of PC3, DU145 and LNCaP cells treated with 0.1 mM AuNPs (AuNP_sp_-PEG, AuNP_st_-PEG and AuNP_r_-PEG) during 24 h. The control were cells without treatment. The data was obtained with the CytExpert 2.3 software, and shown as the mean ± SD.

**Figure 5 cells-12-00787-f005:**
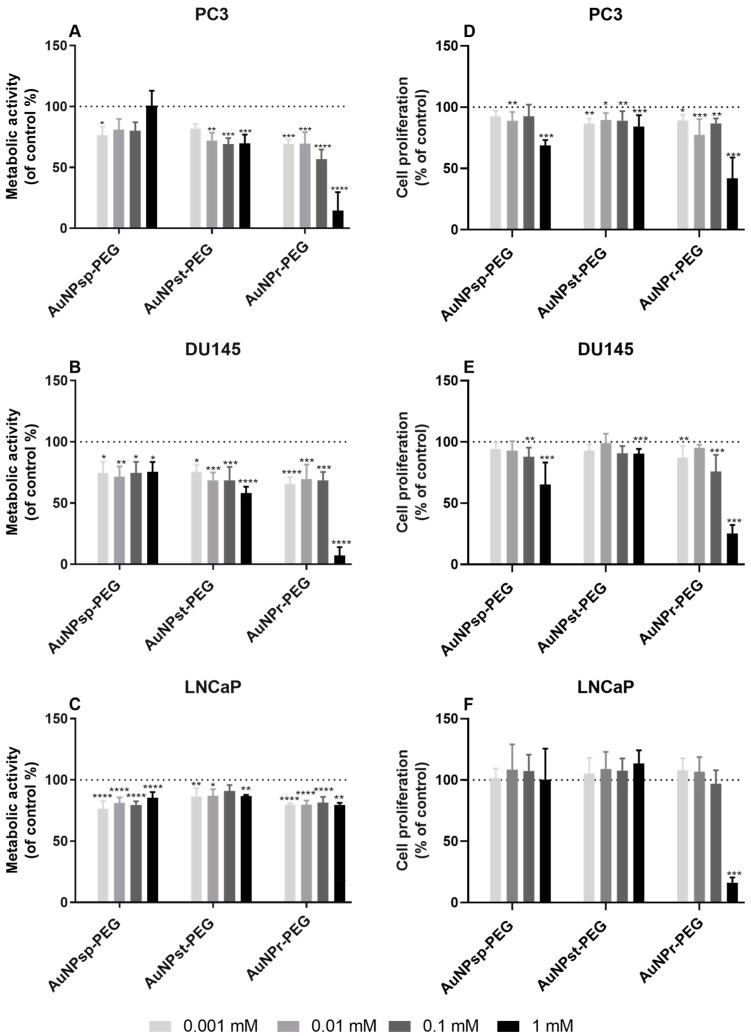
Metabolic activity in PC3 cell line (**A**); DU145 cell line (**B**), and LNCaP (**C**) cell lines. The proliferation of PC3 cell line (**D**); DU145 cell line (**E**), and LNCaP cell line(**F**). (**A**–**F**) Cells were treated with AuNP_sp_-PEG, AuNP_st_-PEG and AuNP_r_-PEG at 0.001, 0.01, 0.1, and 1 mM concentration for 24 h. The results were expressed as a percentage of the control (cells without treatment). Results were presented as mean ± standard deviation; n = 9. Statistical significance was considered as (*) *p* < 0.05, (**) *p* < 0.01, (***) *p* < 0.001, and (****) *p* < 0.0001 when compared to the control.

**Figure 6 cells-12-00787-f006:**
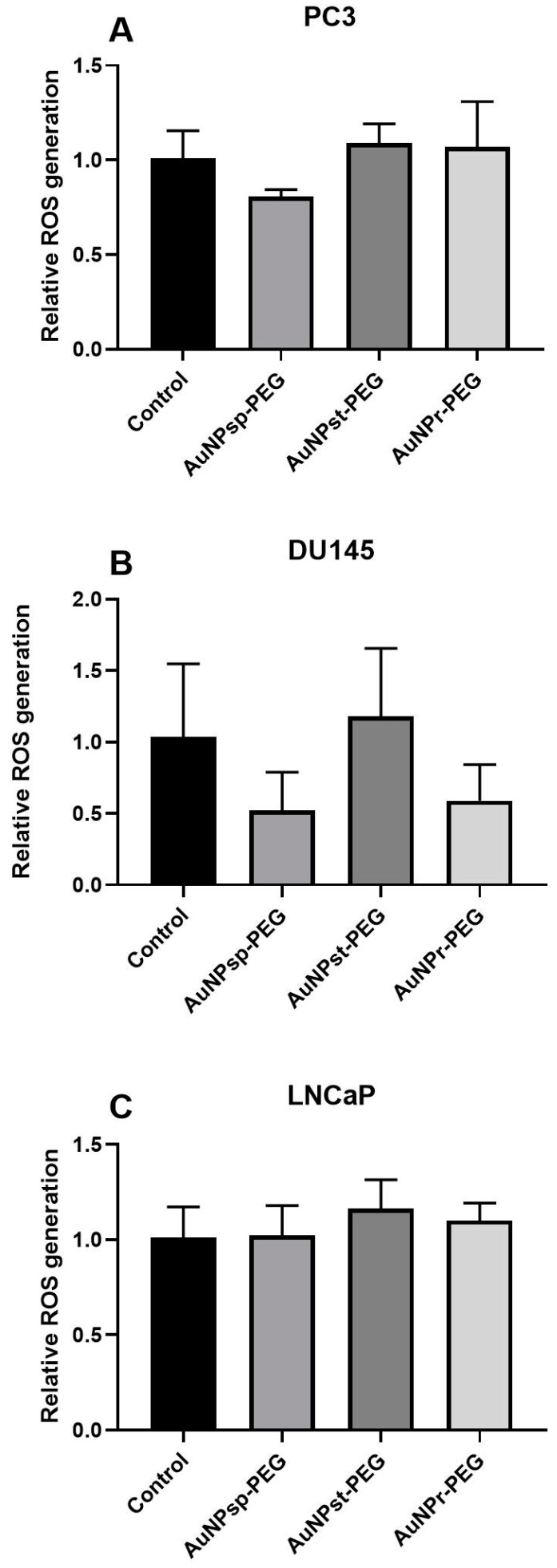
Different types of AuNPs-PEG in PC3 (**A**), DU145 (**B**), And LNCaP (**C**) in ROS levels. Cells were pretreated with 10 µM H2DCFDA for 45 min, then cells were treated with 0.1 mM AuNPs-PEG of different shapes during 24 h. Results represent the mean of DCF fluorescence (mean ± SD) from six independent replicates (n = 6).

**Figure 7 cells-12-00787-f007:**
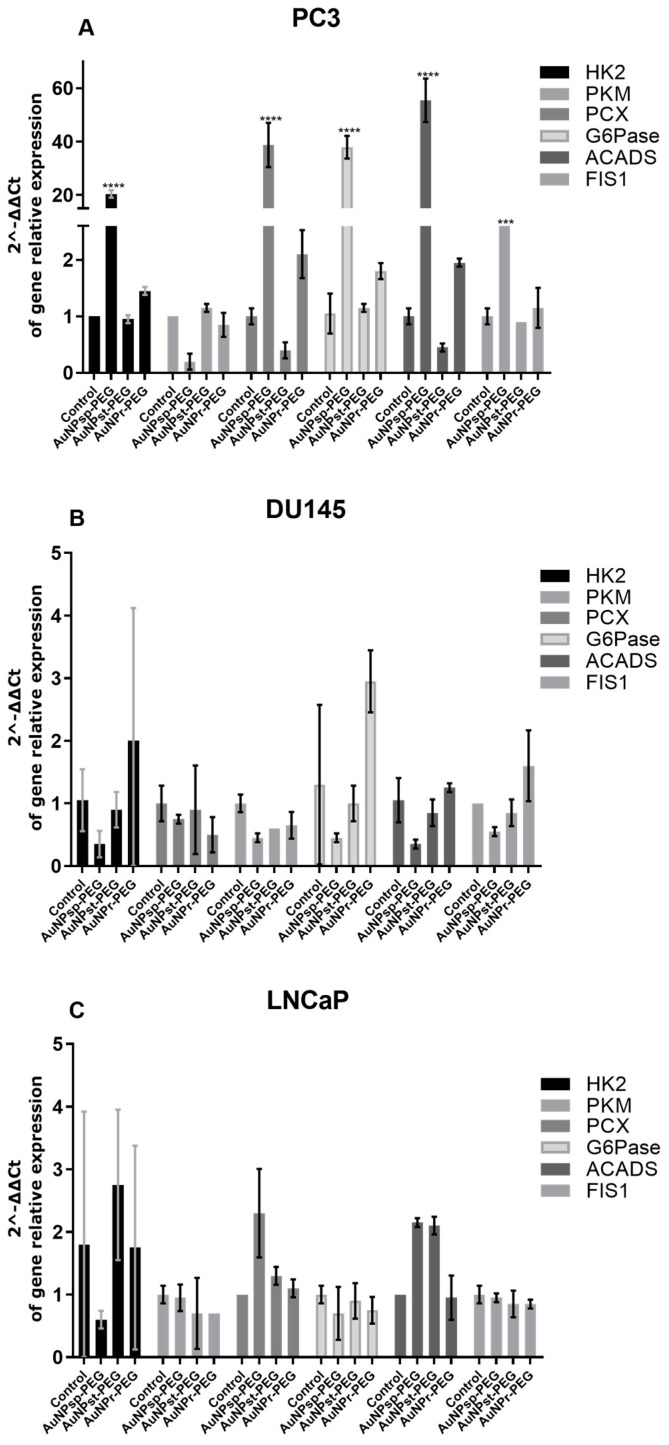
RT-qPCR analysis of genes transcripts involved in different metabolic pathways—hexokinase-2 (HK2) and pyruvate kinase (PKM) on glycolysis, glucose-6-phosphatase (G6Pase) and pyruvate carboxylase (PCX) on gluconeogenesis, acyl-CoA dehydrogenase (ACADS) on beta-oxidation and mitochondrial fission 1 protein (FIS1) after AuNPs-PEG treatment. PC3 (**A**), DU145 (**B**) and LNCaP (**C**). The mRNA expression level of each enzyme was normalized to GAPDH housekeeping gene. Gene relative expression was employed by the ΔCT expression/ΔCT control ratio. (n = 2). Data are shown as means ± SD. Statistical significance was considered as (***) *p* < 0.001, and (****) *p* < 0.0001.

**Figure 8 cells-12-00787-f008:**
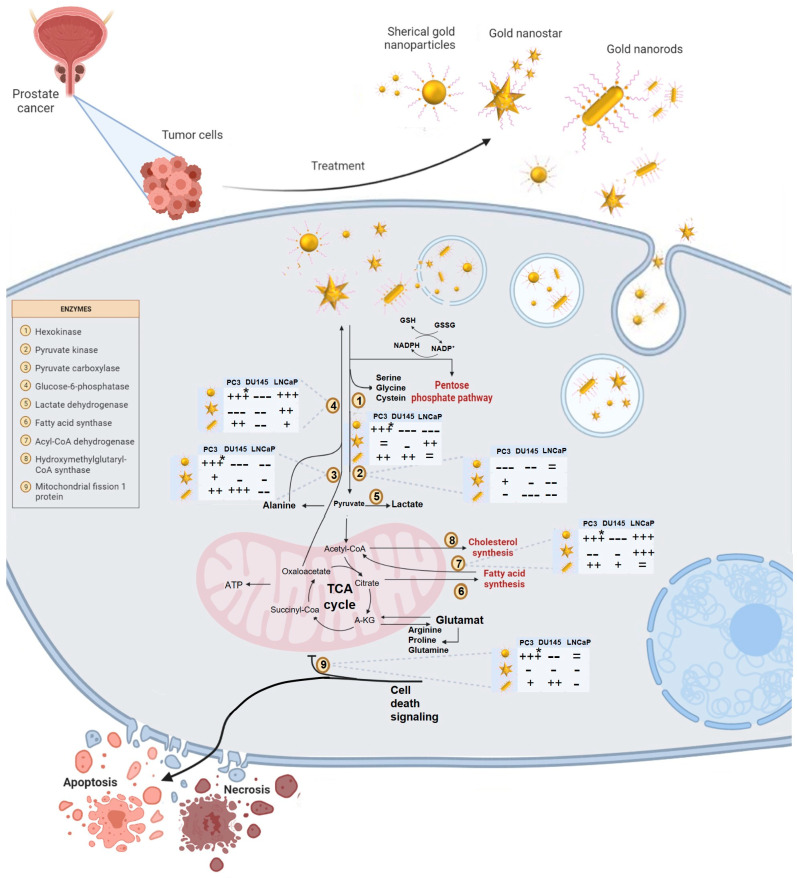
Effect of different types of gold nanoparticles (AuNPs) in several prostate cancer cell metabolic pathways, including glycolysis, gluconeogenesis, and beta-oxidation after internalization. Prostate cancer cells exhibit increased lipid metabolism, which results in citrate synthesis at the tricarboxylic acid (TCA) cycle that is not only an energy source but also for other biomolecules synthesis. Additionally, certain steps of gluconeogenesis seem to be activated to maintain normal glucose levels which can then be used for anabolic purposes. Hexokinase-2, pyruvate kinase, glucose-6-phosphatase, pyruvate carboxylase, acyl-CoA dehydrogenase and mitochondrial fission 1 protein were evaluated by RT-qPCR. The influence of AuNPs-PEG in the expression of metabolic enzymes assessed using +, − or = symbols (+++ > ++ > + and −−− > −− > −). (*) *p* < 0.05.

**Table 1 cells-12-00787-t001:** Set of primer sequences for the one-step multiplex RT-qPCR.

Gene	Primer Forward	Primer Reverse
HK2	GGCAATGAAACCAAAGCCAG	CAAACTAAAAACTCCCCCTTCC
G6Pase	CTCCTCTATCACATTACATCATCC	GAAACATACAAAAGCACCACC
PKM	ATTCACCACCCATCACAGCC	CAGACGAGCCACATTCATTCC
PCX	CATCCCCAACATCCCTTTCC	CCACTTCACAGAACTTGAAGAC
ACADS	AGTGTCAACAACTCTCTCTACC	AAGCAGCCAATTTTGTCACC
FIS1	TGTCCTTTCCCTGTTCTCC	AGCCCCGTTTTATTTACACTC
GAPDH	TCAAGATCATCAGCAATGCC	TGAGTCCTTCCACGATACC

**Table 2 cells-12-00787-t002:** Hydrodynamic diameters and zeta potential of AuNP_sp_-PEG, AuNP_st_-PEG and AuNP_r_-PEG.

Sample	Hydrodynamic Diameter (nm)	Polydispersity Index (PDI)	Zeta Potential (mV)
AuNP_sp_-PEG	47.31 ± 0.46	0.3 ± 0.01	6.7 ± 7.9
AuNP_st_-PEG	109.61 ± 1.27	0.14± 0.01	33.1 ± 12.0
AuNP_r_-PEG	54.58 ± 0.34	0.45 ± 0.01	11.0 ± 18.9

## Data Availability

All data supporting this study’s findings are available within the article or from the corresponding authors upon reasonable request.

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
