# Peer review of "Metabolic Disruption of Gold Nanospheres, Nanostars and Nanorods in Human Metastatic Prostate Cancer Cells"

_cells, 2023, doi:10.3390/cells12050787_

Round 1

Reviewer 1 Report

The authors synthesized gold nanoparticles of different structures and analyzed the different morphologies influence the behavior of cells. The manuscript is well written with full of details and the discussion and conclusion is supported by the experimental results. The manuscript could be accepted by Cells. My only concern is about the style of the Abstract. Please check the author guide and other published papers in Cells.

Author Response

The authors synthesized gold nanoparticles of different structures and analyzed the different morphologies influence the behavior of cells. The manuscript is well written with full of details and the discussion and conclusion is supported by the experimental results. The manuscript could be accepted by Cells. My only concern is about the style of the Abstract. Please check the author guide and other published papers in Cells.

A1: The author's acknowledgement of the Review 1 suggestion. We removed the headings
according to the manuscript preparation guidelines regarding the abstract style.
NOTE: Yellow – English editing ; Green – Suggested alterations made by the authors

Reviewer 2 Report

Dear Authors,

thank you for your manuscript. It's topic is quite urgent, nevertheless, the manuscript contains some serious flaws and need to be carefully revised to be suitable for publication.

1. Synthesis of AuNPs is well-known since XIX century (e.g. Faradey's Au colloids) and the cancer therapy using Au NPs is established in the middle of XX century, see for example https://pubs.acs.org/doi/10.1021/acs.molpharmaceut.8b00810. Therefore the Authors' statement "Gold nanoparticles (AuNPs) are a novel type of nanoparticles..."  raises questions about the novelty of this study. I suppose the actuality has to be strengthened to clarify this issue for the readers.

2. The manuscript contains too much typos and incomprehensible language constructs, e.g.

Line 55, "and quickly functionalizes with many (bio)molecules";

Line 74, "some patients continuous do not respond";

Line 77, "overcome AuNPs lack of knowledge of prostate cancer";

Line 103, "were prepared by Turkevich method and his co-workers";

Line 182, "Effect AuNPs in cell structure by TEM";

Line 230, "The 260/280 nm ratio were used";

Line 255, "Different surface plasmon resonance";

Line 258, "Scheme of sphere gold nanoparticles",

Line 326, "found in the three metastatic cell lines analysed", etc.

The language of the manuscript has to be improved with by the native-speaker.

3. The Figures' quality has to be improved, since the text in Figures 1, 2, 5, 8 is unreadable. The scale bar in Figure 3 is too small.

4. All the abbreviations have to be introduced when firstly used, e.g. UV, TEM, SEM, DLS.

5.  The text contains some fragments that do not carry a semantic load and are not necessary, from my point of view, in a scientific article, e.g. "Authors should discuss the results and how they can be interpreted from the perspective of previous studies and of the working hypotheses. The findings and their implications should be discussed in the broadest context possible. Future research directions may also be highlighted. AuNPs were characterized by distinct methods and the mean size of AuNPs was measured by TEM and DLS and the shape was confirmed by UV-Vis spectra, TEM and SEM image analysis";

"...concern because the principal aim of this 613 nanomaterial is to be applied in the field of biomedical to increase the safety of AuNPs 614 application".

6. The Authors state the aspect ratio of the Au NPs cannot be estimated using dynamic light scattering techniques. I cannot completely agree with this, since there is a depolarized dynamic light scattering method allowing the analysis of NPs shape, e.g. https://onlinelibrary.wiley.com/doi/abs/10.1002/ppsc.201800388, https://www.mdpi.com/2076-3417/11/1/183, https://pubs.rsc.org/en/content/articlelanding/2016/nr/c6nr03386e

7. The Conclusion section needs to be supplemented with the additional detailed description of what the authors have brought new to such a well-developed topic as gold nanoparticles, so that the reader can evaluate the usefulness of this work for basic science and/or applied medicine. 

Round 2

Reviewer 2 Report

Dear Authors,

Thank you for careful addressing all my comments. I believe now the manuscript can be accepted for publication.